# Host Innate and Adaptive Immunity Against African Swine Fever Virus Infection

**DOI:** 10.3390/vaccines12111278

**Published:** 2024-11-13

**Authors:** Tianqi Zhang, Zixun Lu, Jia Liu, Yang Tao, Youhui Si, Jing Ye, Shengbo Cao, Bibo Zhu

**Affiliations:** 1National Key Laboratory of Agricultural Microbiology, Hubei Hongshan Laboratory, College of Veterinary Medicine, Huazhong Agricultural University, Wuhan 430070, China; tianqi@webmail.hzau.edu.cn (T.Z.); luzixun@webmail.hzau.edu.cn (Z.L.); liujia000808@163.com (J.L.); taoyang0802@webmail.hzau.edu.cn (Y.T.); youhui@mail.hzau.edu.cn (Y.S.); yej@mail.hzau.edu.cn (J.Y.); sbcao@mail.hzau.edu.cn (S.C.); 2Frontiers Science Center for Animal Breeding and Sustainable Production, Huazhong Agricultural University, Wuhan 430070, China; 3The Cooperative Innovation Center for Sustainable Pig Production, Huazhong Agricultural University, Wuhan 430070, China

**Keywords:** African swine fever, ASFV, innate immunity, adaptive immunity, immune evasion

## Abstract

Africa swine fever virus (ASFV) is the causative agent of African swine fever (ASF), a highly contagious hemorrhagic disease that can result in up to 100% lethality in both wild and domestic swine, regardless of breed or age. The ongoing ASF pandemic poses significant threats to the pork industry and food security, with serious implications for the sanitary and socioeconomic system. Due to the limited understanding of ASFV pathogenesis and immune protection mechanisms, there are currently no safe and effective vaccines or specific treatments available, complicating efforts for prevention and control. This review summarizes the current understanding of the intricate interplay between ASFV and the host immune system, encompassing both innate and adaptive immune responses to ASFV infection, as well as insights into ASFV pathogenesis and immunosuppression. We aim to provide comprehensive information to support fundamental research on ASFV, highlighting existing gaps and suggesting future research directions. This work may serve as a theoretical foundation for the rational design of protective vaccines against this devastating viral disease.

## 1. Introduction

ASF is an acute, hemorrhagic, and fatal viral disease affecting domestic pigs and wild boars, causing significant economic losses for the global swine industry. Despite decades of research, the complex nature of ASFV, including its genetic diversity and pathogenic mechanisms, has impeded the development of effective prevention and treatment strategies. Originating in Kenya in the early 20th century, ASFVs circulated in many African countries. They can be classified into 24 genotypes (I-XXIV), all of which can be detected across Africa [1,2,3]. Genotype I was first introduced into Europe in 1957, leading to outbreaks in many European countries [4]. This genotype was eradicated in Europe by the mid-1990s [5]. In 2007, the highly virulent genotype II (Georgia07) was isolated in Georgia and rapidly spread throughout the Caucasus region, eventually reaching the Russian Federation and Eastern Europe [6]. By 2018, a Georgia07-like ASFV (Pig/HLJ/18) had spread to China and other Asian countries [7,8]. In addition, genotype I also circulates in China [9]. Currently, genotype II is widely circulating among pig populations in Europe, Africa, and Asia [8,10], highlighting the severe threat ASF poses to global food security. As ASFV continues to spread and evolve, there is great concern about recombinant and/or new virulent variants that may evade pre-existing immunity, resulting in a major challenge for disease control and vaccine development [9].

ASFV infection in domestic pigs exhibits diverse clinical manifestations, including subacute, acute, and hyperacute forms, with outcomes influenced by the virulence of the viral strain and the host’s immune status [5]. Acute ASF presents with hallmark signs such as elevated body temperature, lethargy, loss of appetite, reduced activity, respiratory difficulty, and pronounced pulmonary edema [11]. Subacute ASF, typically associated with less virulent strains, exhibits similar but milder clinical signs [12]. While acute and hyperacute forms have high mortality rates, subacute cases can lead to mortality rates of 30% to 70% [13]. Despite extensive efforts over the past century, no approved vaccines or antiviral treatments for ASFV infection exist globally. Current prevention and control measures rely on early detection through effective laboratory diagnosis, the quarantining and culling of infected pigs, and rigorous sanitary measures, underscoring the urgent need for an ASFV vaccine [14,15].

ASFV is classified under the genus *Asfivirus*, the sole member of the *Asfarviridae* family [16]. Its genome is a linear double-stranded DNA molecule ranging from 170 to 190 kb in length, with significant variations in length, and with significant variations attributed to the presence of different multigene family (MGF) members [17,18,19]. The genome encodes between 150 and 200 open reading frames, specifying 54 structural proteins of the virion particle and over 100 proteins involved in infection [20]. These proteins play roles not only in viral replication and assembly but also in modulating host immune responses [21,22]. However, more than half of those proteins have no known or predicted functions [20]. Notably, ASFV primarily targets and replicates within mononuclear phagocytic cells (macrophages and monocytes), though infection is not restricted to specific cell types [23,24]. The invasion of monocytes and macrophages triggers both innate and adaptive host responses; however, ASFV has evolved mechanisms to manipulate host immunity, including the regulation of interferon (IFN) signaling, the inhibition of autophagy and apoptosis, and the disruption of antigen presentation and lymphocyte activation [23,24]. This review will explore the host innate and adaptive immune responses to ASFV infection, aiming to provide insights for future vaccine development efforts.

## 2. Innate Immune Responses

Upon viral infection, the innate immune system acts as the body’s initial defense mechanism. Most innate immune cells are equipped with mechanisms to recognize and respond to viral infections via pattern recognition receptors (PRRs), which detect conserved pathogen-associated molecular patterns (PAMPs) or other molecular signatures of pathogens [25]. Viral activation of PRRs in these cells stimulates the production and release of type I interferon (IFN-I), type III interferon (IFN-III), and various pro-inflammatory mediators, including cytokines, chemokines, and antimicrobial peptides. These responses aid in restricting the virus’s spread and triggering the activation of the host’s adaptive immune response [26]. Consequently, the intrinsic responses of innate immune cells establish a cellular environment conducive to a subsequent antiviral state in the host.

### 2.1. Macrophages

Macrophages are evolutionarily conserved tissue-resident or infiltrated immune cells that belong to the mononuclear phagocyte system. They present in nearly every tissue and play critical roles in tissue development, homeostasis, and remodeling, as well as the repair of damaged tissue. In response to virus infection, macrophages are among the first immune cells to encounter viral PAMPs, initiating a range of antiviral and inflammatory responses. In addition, they phagocytose local cellular debris and antibody-opsonized particles via the Fc receptor, thereby preventing viral dissemination [27,28]. Macrophages are the primary target cells for ASFV infection, where the viral genome undergoes replication and facilitates the assembly of virion components, leading to the release of progeny virions [23]. Therefore, the interaction of ASFV and macrophages is of major importance that determines immunopathogenesis of the virus. In response to ASFV infection, the PRRs within macrophages recognize the virus, triggering IFNs and inflammatory responses (Figure 1). These processes contribute to the establishment of an “antiviral state” in both infected and neighboring cells [29].

In an in vitro study utilizing single-cell RNA sequencing, the comprehensive transcriptome landscape of ASFV-infected primary porcine alveolar macrophages is analyzed, revealing that most upregulated genes in viral exposed cells are associated with innate immune signaling pathways. Notably, IFN-stimulated genes (ISGs) such as IFI6, ISG12, and MX1, along with inflammatory and cytokine-related genes, are significantly increased in exposed cells, suggesting the functional gene activation involved in antiviral signaling pathways aimed at combating the infection [30]. Consistently, an in vivo single-cell RNA sequencing analysis of macrophages from the spleens of ASFV-infected pigs also shows moderate levels of ISGs at day 5 post infection, with expression decreasing by day 7. Importantly, ISG levels are significantly decreased in infected cells compared to bystander cells, indicating that ASFV can suppress IFN-mediated antiviral responses in infected macrophages [31]. Additionally, analysis of RNA sequencing data from porcine macrophages infected with ASFV-CN/GS/2018 at various time points in vitro indicates an upregulation of certain antiviral and inflammatory factors [29]. The majority of differentially expressed genes initially display elevated expression at 4 h post infection, followed by reduced expression at 16 h post infection, relative to mock-infected cells, as identified through microarray analysis of macrophages infected with a virulent ASFV strain. These genes include pro-inflammatory cytokines such as TNF-α, IL-6, and IFN-β, as well as chemokines from the CC and CXC families, all of which are part of the host’s response to eliminate viral infection. The return to baseline expression of these upregulated genes at later time points may be attributed to the inhibitory effects of ASFV-encoded proteins on the host’s innate immune response [32]. Furthermore, the cytokine profiles in ASFV-infected macrophages may vary depending on strain virulence. For instance, infection with wild-type ASFV (Pr4) or an MGF360/530 deletion mutant ASFV (Pr4Δ35) results in distinct transcriptional profiles for macrophage cytokines. The transcriptional patterns observed in Pr4Δ35-infected cells are primarily indicative of a type I IFN response or a combined response to IFN, double-stranded RNA (dsRNA), and/or viral infection, unlike the response to wild-type ASFV. This suggests that the MGF360/530 proteins are responsible for inhibiting IFN responses [33]. Additionally, the cyclic GMP-AMP synthase (cGAS), an important DNA sensor in macrophages, can detect dsDNA from attenuated ASFV strains (e.g., NH/P68) but not from virulent strains (e.g., Armenia/07), resulting in the production of IFN-I through activation of the STING, TBK1, and IRF3 signaling pathways [34]. Thus, the cGAS-STING-IRF3 pathway plays an essential role in the host response to ASFV infection, where production or inhibition of IFN-I varies depending on whether the strain is attenuated or virulent. This underscores the notion that ASFV virulence influences virus-mediated modulation of the innate immune response [34]. Macrophages infected with attenuated ASFV strains may secrete pro-inflammatory cytokines and IFNs that could potentially enhance immune surveillance and facilitate the development of appropriate adaptive immune responses.

Macrophages represent highly heterogeneous populations with remarkable diversity and plasticity. In response to various environmental stimuli or different pathophysiologic conditions, macrophages can modify their phenotype and function via distinct phenotypic polarization [35]. The two extremes of this dynamic state are represented by classically activated (M1) and alternatively activated (M2) macrophages, reflecting the Th1–Th2 polarization seen in T cells. The M1 phenotype is stimulated by pro-inflammatory cytokines or microbial components (e.g., IFN-γ, TNF, and LPS), exhibiting antimicrobial and pro-inflammatory functions. Conversely, the M2 phenotype, which is often referred to as the “resting” state, is enhanced by IL-13, IL-4, or IL-10 in the absence of infection and is associated with immunosuppression and wound healing [36]. M1 polarization in swine, which is similar in other species, could be induced in vitro by LPS and IFN-γ, leading to the production of pro-inflammatory cytokines. M2 polarization, induced by IL-4, is marked by the upregulation of Arginase 1 (Arg-1) and CD203a [37]. To date, most studies have focused on ASFV infection in non-polarized macrophages, creating a knowledge gap regarding the interactions between ASFV and polarized macrophages. One study examines the interactions between porcine monocyte-derived M0, M1, and M2 macrophages and ASFV strains of diverse virulence in vitro. It finds that M1 macrophages infected with the attenuated NH/P68 strain produce elevated levels of IL-1α, IL-1β, and IL-18 compared to those infected with the virulent 22653/14 strain. While both strains efficiently replicate in the macrophage subsets tested, the NH/P68 strain shows reduced infectivity in M1 and M0 macrophages activated with IFN-α relative to unactivated M0 macrophages. Notably, M2 polarization does not significantly alter macrophage responses or susceptibility to ASFV infection [38]. However, in vitro studies show that the induction of monocyte-derived macrophages (moMΦ) toward pro-inflammatory or anti-inflammatory phenotypes using TLR2 agonists or IL-10/TGF-β does not significantly affect susceptibility to ASFV. However, TLR2 activation does enhance the replication of the 26544/0G10 strain during low-virulence infections, potentially linked to an increase in IL-1Ra production. CD163, which is associated with the phagocytic capacity of moMΦ and porcine monocyte-derived DCs (moDCs) during ASFV infection, does not exhibit altered expression following pre-treatment with IL-10 [39,40]. In another study, M2 macrophages exhibit slightly higher permissiveness to the replication of the avirulent BA71V strain compared to M0 and M1 macrophages [41]. Meanwhile, ASFV infection in porcine alveolar macrophages led to enhanced levels of the M2 polarization marker Arg-1 [37,42], which promotes ASFV replication [42]. These findings suggest that ASFV may induce M2 polarization to evade cellular defense mechanisms, potentially increasing susceptibility to attenuated ASFV strains. Therefore, macrophages in a distinct polarization likely exhibit varied responses to both attenuated and virulent strains. Further insights into the virological and cellular factors associated with the dynamic changes in macrophage polarization during ASFV infection are essential to clarify the molecular mechanisms underlying disease progression and to develop innovative macrophage-based therapies against ASFV.

### 2.2. Dendritic Cells

Dendritic cells (DCs) act as the sentinels of the host immune system, liking innate immune responses and molecular sensors with the activation of adaptive immunity. As key professional antigen-presenting cells (APCs), DCs perform several crucial functions, including the recognition and acquisition of antigens, followed by migration to regional lymph nodes for presenting processed antigens to lymphocytes, ultimately inducing CD4+ or CD8+ T-cell responses [43]. Due to their lower frequency in the blood and tissues of pigs, monocytes cultured in vitro can differentiate into DCs in media supplemented with recombinant GM-CSF and IL-4 [44]. Despite the essential role of DCs in bridging innate and adaptive immunity, the number of studies investigating the response to ASFV is limited. One study characterizes the interactions between moDCs and ASFVs with strains of distinct virulence in vitro. All strains are found to replicate efficiently in immature moDCs; however, maturation with IFN-α reduces the susceptibility of moDCs to low-virulence ASFV. Notably, infection with attenuated strains, but not virulent isolates, downregulates expression of SLA I (the swine MHC-I surface protein) on infected moDCs, potentially impairing the effective initiation of adaptive immune response [45]. Additionally, another study evaluates the response of blood DCs to ASFV infection. Enriched porcine leukocytes produce high levels of IFN-I after ASFV infection in vitro, suggesting that plasmacytoid DCs (pDCs) may be a significant source of IFN-I in pigs infected with ASF [46]. The ASFV late viral protein p72 has been identified within DCs in multiple organs of swine and wild boars infected with the moderately virulent strain [47]. However, further investigation is needed to determine whether ASFV can directly infect DCs. Future research may focus on the interactions between ASFV and distinct subsets of bona fide DCs in vivo, taking into account the diverse virulence of the ASFV strains involved.

### 2.3. Natural Killer Cells

Natural Killer (NK) cells are integral parts of the innate immune system and belong to the family of innate lymphoid cells. They play crucial roles in early antiviral responses by producing effector cytokines, eliminating cells infected with viruses, and facilitating the adaptive immune response [48]. Given their importance in viral clearance, NK cells are thought to be significant to developing an effective immune response to ASFV. Indeed, robust IFN-γ responses have been observed in NK cells at day 14 post infection with the ASFV vaccine strain in swine, with the response remaining robust following challenge [49]. Furthermore, one study shows that early NK cell activation is positively correlated with protection induced in vivo by the attenuated NH/P68 isolate in domestic pigs. The pigs that survived displayed protection against subsequent challenges with the homologous virulent ASFV strain L60 [50]. Therefore, NK cells may play a role in shaping antiviral immunity and defense against infection, with their efficacy potentially modulated by the virulence of the ASFV strain employed.

### 2.4. Innate Immune Evasion

To replicate and spread within host cells, primarily mononuclear phagocytic cells, ASFV relies on highly specific interactions between viral components and the infected cells. This interaction brings up the subversion of multiple cellular signaling pathways that regulate various cellular functions. Consequently, ASFV has matured mechanisms to evade immune responses by encoding a range of immune evasion proteins, containing those that inhibit IFN expression, regulate autophagy and apoptosis, and impair antigen presentation by APCs (Figure 1) [23,51,52]. For instance, members of the MGF505 and MGF360 families are linked with ASFV virulence and target critical molecules in the RIG-I and cGAS-STING signaling pathways, influencing IFN-I induction in swine macrophages [53,54,55,56]. In addition to these MGF proteins, other ASFV-encoded proteins, such as pE184L, pC129R, pDP96R, and pL83L, impair the activation of the cGAS-STING-TBK1 signaling pathways, resulting in the negative regulation of IFN-I production [57,58,59,60]. Furthermore, ASFV-encoded proteins CD2v and pH240R inhibit the IFN-JAK-STAT axis via the interaction with IFNAR1 and IFNAR2, which inhibits the expression of ISGs, leading to the impairment of host antiviral effects [61,62]. Notably, ASFV has acquired numerous genes that regulate NF-κB- and NLRP3-mediated inflammatory responses. For example, ASFV-encoded pF317L affects the phosphorylation of IKKβ, which in turn inhibits the phosphorylation and ubiquitination of the IκBα molecule, thereby reducing the activation of the NF-κB signaling pathway [63]. Additionally, ASFV proteins pMGF505-7R and pH240R interact with NLRP3 to prevent ASC oligomerization, leading to decreased NLRP3 inflammasome formation, thereby decreasing caspase 1 activation and IL-1β production [64,65]. The ASFV I329L protein can suppress TLR activation, thereby weakening the immune response of macrophages [66]. Certain ASFV-encoded proteins can also activate autophagic degradation, promoting the autophagy-mediated breakdown of critical components in the cGAS-STING signaling pathway and regulating innate immune responses. For example, the pL83L protein interacts with cGAS and STING to inhibit IFN-I production by facilitating the autolysosomal degradation of STING through the recruitment of Tollip [60]. ASFV protein p17 induces mitophagy by enhancing the interaction between the mitophagy receptor SQSTM1 and TOMM70. This process leads to the degradation of antiviral signaling proteins in mitochondria, ultimately inhibiting the production of inflammatory cytokines and IFN-I [67]. Additionally, ASFV protein CD2v interacts with CSF2RA, belongs to the hematopoietic receptor superfamily found in myeloid cells, and is a crucial receptor for activating JAK and STAT proteins. This interaction regulates the JAK2-STAT3 signaling pathway and inhibits apoptosis, thereby promoting ASFV replication [68].

MHC class I and II molecules—referred to as SLA class I and II in swine—process and present antigens on the cell surface to CD8+ and CD4+ T cells, respectively. ASFV may disrupt antigen presentation by modulation of SLA class I and II expression in infecting monocytes, macrophages, and DCs, thus compromising the host immune system. Notably, the diverse virulence of ASFV strains differently affects SLA class I expression while SLA class II expression remains unchanged [41,69]. For instance, porcine monocyte-derived macrophages infected with the virulent ASFV strain induce SLA class I expression [70]. In contrast, monocyte-derived macrophages and DCs infected with attenuated genotype I ASFV strains downregulate SLA class I expression, whereas virulent genotype I strains do not have this effect [41,45], suggesting the modulation of SLA class I expression by ASFV is strain-dependent. Additionally, ASFV has been demonstrated to suppress the immune response by obstructing antigen binding to SLA-II, even though its expression remains stable [41,71]. Consequently, ASFV may impair early antigen presentation by APCs through its effects on both SLA class I and II, leading to the suboptimal development of the acquired immune response. Furthermore, ASFV infection significantly decreases CD16 expression on the surface of macrophages and monocytes, which may impair their antiviral activity and overall function [41,69]. Collectively, these findings indicate that the ongoing interaction between ASFV and host cells undermines the function of macrophages or DCs, inhibiting the initiation of virus-specific immune responses.

## 3. Adaptive Immune Responses

Although innate immune responses are crucial for controlling viral replication during the early stages of infection, complete viral clearance—achieved by preventing the generation of new virions and eliminating infectious virions—necessitates the activation of adaptive immune responses. While innate immune responses are rapidly triggered by PAMPs derived from viral infection, the adaptive immune response usually takes days to weeks to develop [43]. The adaptive immune response encompasses the coordinated activities of humoral and cellular immunity, mediated by adaptive immune cells, including B and T cells. B cells are responsible for the generation of antibodies that neutralize pathogens and enhance the functions of innate immune cells. CD4+ T cells facilitate the activation of both innate and adaptive immune responses, which can aid cytolytic activity and B-cell function. In contrast, CD8+ T cells could directly eliminate infected cells. Additionally, unconventional T cells play a significant role in initiating both antiviral responses and adaptive immune responses (Figure 2) [26].

### 3.1. B-Cell-Mediated Responses

B cells support virus clearance mainly by generating specific antibodies against the virus, including both neutralizing and non-neutralizing antibodies. Neutralizing antibodies interact with viral surface proteins, preventing free virions from infecting susceptible uninfected cells and thereby controlling viral dissemination. They also bind to viral proteins displayed on the surface of infected cells, initiating complement-dependent cytotoxicity (CDC) and antibody-dependent cellular cytotoxicity (ADCC), which result in the lysis or clearance of infected cells [72]. Therefore, the production of these antibodies is a primary objective of many vaccine strategies. During ASFV infection, the role of antibody production in protection is somewhat contradictory [73]. For instance, pigs immunized with the ASFV proteins exhibit a strong neutralizing antibody response, which has been associated with protection against virulent ASFV infection [74]. However, another study shows that, despite the induction of neutralizing antibodies against ASFV proteins, these antibodies alone are insufficient for adequate protection [75]. Pigs surviving ASFV infection demonstrate higher antibody levels, yet clinical signs remain evident [76]. Furthermore, protection against ASFV can occur even in the absence of neutralizing antibodies [77,78], suggesting that cellular immunity is crucial for clearing ASFV infection. Of note, the main ASFV antigens that are recognized for eliciting neutralizing antibodies include p72, p54, p30, and CD2V, which are among the most extensively studied for ASF vaccine development [75,79,80,81,82,83]. Indeed, subunit vaccines composed of proteins have shown partial protection against virulent ASFV challenge [84,85]. Additionally, other immunogens, such as pp62, p15, p22, and EP153R/C-type lectin, have also been suggested to elicit distinct antibody responses [77,86]. Immunogenicity studies with various ASFV antigen cocktails have been conducted in swine, revealing antigen-specific antibody responses against p17, p15, and EP153R/C-type lectin. However, these responses do not exhibit neutralizing activity despite there being specific neutralizing antibodies against p72, p30, and p54 present [86,87,88].

In addition to the protective roles of antibodies in ASFV infection, antibody-dependent enhancement (ADE) of infection has been found in vaccine immunization studies. For instance, pigs that have been immunized with a DNA vaccine exhibit a broad antigen-specific antibody response; however, these pigs are not protected and show higher viremia three days post ASFV challenge. Moreover, immune sera from these vaccinated animals lack neutralizing activity and appear to enhance macrophage infection in vitro [89,90]. Similarly, pigs vaccinated with a combination of ASFV DNA, proteins, or inactivated virus exhibit elevated antibody titers that lack the ability to neutralize the virus, correlating with increased infection and a more severe disease course [91,92]. These results indicate that excessive antibody production may be harmful, potentially hastening disease progression through ADE of infection. This process may be mediated by IgG antibody and Fc receptor signaling [73], although the underlying mechanisms require further investigation. Numerous vaccine strategies for ASFV have been explored, and various ASFV antigens have been evaluated; however, many candidates have yet to be investigated against virulent virus challenges. Consequently, it remains unclear which ASFV antigens significantly contribute to protection against virulent ASFV or are associated with immune enhancement in ASFV pathogenesis. Thus, a comprehensive understanding of the functions of individual ASFV proteins is essential for the rational design of safe and effective vaccines.

### 3.2. T-Cell-Mediated Responses

Antiviral T-cell responses are initiated when naïve virus-specific T cells recognize peptides from viruses presented by MHC class I and II molecules on APCs. Upon activation, these T cells proliferate and differentiate into effector T cells, orchestrating antiviral responses at the infection site [26]. Most current literature focuses on T cell responses in mice and humans, with comparatively few studies examining porcine T-cell phenotypes and functions at homeostasis or during virus infection. Traditional porcine T cells consist of three subsets: CD4+ T helper cells, CD8+ T cells, and CD4+ CD8α+ double-positive (DP) T cells, which differs from the T cell subsets found in mice and humans [24,93]. Following a highly virulent ASFV infection, cellular responses in domestic pigs are predominantly characterized by an increase in DP T cells within lymphatic organs and the peripheral blood, with a concomitant decrease in CD4+ and CD8 + T-cell frequency. These DP T cells show less proliferative activity and lower perforin expression [94,95]. In contrast, after moderately virulent ASFV infection, the frequencies of DP T and CD8+ T cells increase in the spleens, lungs, and livers of swine, with DP T cells showing proliferation primarily in the spleens, but not in other tissues [96]. However, the involvement of DP T cells in antiviral Th1 response needs further exploration.

Specific T-cell immune responses are quickly activated following ASFV infection but gradually decline in the late stages of infection in surviving pigs, suggesting that T cells play critical roles in mediating antiviral responses [97]. Studies have demonstrated significant upregulation of the lymphocyte activation marker CD69 on T cells, alongside an increased proportion of tetramer+ CD8+ T cells during ASFV infection [76,98]. Moreover, a correlation between protection against ASFV infection and the presence of virus-specific T cells, even in the absence of specific antibodies in animal models, is observed [50,77,78,99]. The inoculation of pigs with live attenuated virus provides protection against lethal challenges with homoeologous virulent ASFV, which is associated with the induction of cytotoxic activity in CD8+ T cells [94,100,101,102]. Notably, the protection conferred by a naturally attenuated strain is abrogated following the depletion of CD8+ lymphocytes by antibodies, highlighting the essential role of CD8+ T cells in providing robust protection against ASFV infection [77,99]. However, the specific ASFV antigens that effectively stimulate T cells, particularly the CD8+ T-cell subset, and induce a strong protective response, remained elusive [82,87]. To determine ASFV T-cell epitopes with protective potential, multiple approaches—including immunopeptidomics, in silico predictions, and antigen presentation assays—are employed, laying the groundwork for subunit vaccine development [103,104]. For instance, a total of 3818 peptides from 165 pools, representing 133 open reading frames, are selected to identify T-cell peptides that elicit a response against ASFV through IFN-γ ELISPOT assays performed on immune lymphocytes. This screening identified p30, pp62, and p72 as significant inducers of IFN-γ responses in ASFV-immunized porcine lymphocytes [82]. Additionally, T-cell epitope regions within CD2v and C-type lectin proteins are implicated in the T-cell immune responses of infected, vaccinated, and surviving pigs. Epitopes from p72 and pp62 proteins elicit the strongest protective immunity among the five ASFV proteins [76,80,105].

Notably, pigs vaccinated with plasmid DNA encoding ASFV antigens or subunit vaccine formulations exhibit an activation and increased CD8+ T-cell numbers in their blood compared to control pigs. However, this immunization only provides partial protection against genotype I ASFV strain isolates and fails to confer any protection against highly virulent strains [77,78,90,106]. To improve the efficacy of DNA vaccines, pigs can be primed with a cocktail of 15 plasmids and subsequently boosted with a suboptimal dose of live attenuated virus followed by a lethal challenge with virulent ASFV strains. Priming with plasmids encoding CD8+ T-cell antigens has been shown to generate a robust T-cell response and enhance protection against ASFV challenge [107]. However, studies have also observed that pigs immunized with a cocktail of multiple dominant ASFV T-cell epitope genes exhibit activation of IFN-γ-expressing T cells but remain unprotected [82,88]. These findings indicate that achieving efficient protection in ASFV-infected pigs through T-cell-mediated immunity is complex and not yet fully understood. Therefore, improving our understanding of ASFV-protective antigens as well as the relevant epitopes will facilitate the design of safe and effective ASFV vaccines, utilizing complex subunit vaccine formulations alongside optimal combinations of expression vectors and immune adjuvants.

### 3.3. γδ T-Cell Response

The γδ T cells represent a unique subset of unconventional T cells, exhibiting both innate and adaptive functions. Unlike traditional CD4+ and CD8+ T cells, γδ T cells recognize nonpeptide antigens and are unrestricted by classical MHC molecules, making them significant contributors to antiviral immunity [108]. In contrast to mice and humans, pigs possess a high proportion of circulating γδ T cells, which account for approximately half of the total peripheral lymphocyte populations in the blood of young pigs [109]. These cells can be classified into three subpopulations based on the expression of CD2 and CD8α, including naïve, activated, and terminally differentiated effector γδ T cells [110,111]. Upon activation, γδ T cells quickly respond to viral infections by secreting a range of cytokines without requiring clonal expansion or differentiation [108,112]. However, the interactions between ASFV and porcine γδ T cells are not well known. One study finds that, in domestic pigs with moderately virulent ASFV infection, increased circulating γδ T-cell frequencies are positively correlated with survival [113]. Conversely, during infections with highly virulent ASFV, γδ T-cell frequencies are impaired and do not correlate with the survival of infected pigs [95]. This discrepancy may be due to the differing virulence of ASFV strains. In pigs vaccinated with attenuated ASFV, the antigen-specific responsiveness of γδ T cells is assessed via IFNγ production, revealing an enhanced proportion of these cells producing IFNγ by 7 days post infection, an early response compared to traditional T cells and NK cells [49]. Additionally, γδ T cells can function as professional APCs [114], as evidenced by their capacity to present viral antigens to specific T cells in immune pigs [115]. However, the detailed functional roles of these cells remain largely unexplored. Given their high frequency in pigs, γδ T cells may play a role in ASFV infection, warranting further investigation into their significance.

### 3.4. Adaptive Immune Evasion

The disruption of both humoral and cellular components of the host’s adaptive immune responses is a primary strategy employed by ASFV to evade immune detection and elimination. Following ASFV infection, the percentages of total B cells and CD21+ B cells are reduced [116]. As CD21 is a key marker of naïve B-cell maturation and activation, as well as a component in complement activation, these observations suggest impaired B-cell development, leading to an inadequate humoral response [116]. Although certain studies demonstrate the successful induction of neutralizing antibodies against viral proteins, these antibodies fail to confer sufficient protection [75]. This phenomenon may be explained by the production of apoptotic bodies (ApoBD) in ASFV-infected macrophages, which mediates the virus’s evasion of antibody neutralization [117]. Notably, swine serum can inhibit the infection efficiency of intracellular virions, but not the extracellular virions associated with ApoBD, suggesting that ASFV may have hijacked normal cellular pathways to evade humoral responses [117]. Moreover, ASFV infection impairs cellular immune responses by inducing widespread lymphocyte apoptosis, culminating in lymphocytopenia [118]. Huhr et al. demonstrate that domestic pigs infected with highly virulent ASFV strains exhibit reduced counts of CD8+ and CD4+ T cells in peripheral blood [95]. In addition, expression levels of the effector molecule perforin in cytotoxic T cells residing in lymphoid organs are markedly diminished [95,96]. Consistently, ASFV CADC_HN09 strain infection results in progressive lymphocyte apoptosis in peripheral blood over time, predominantly affecting B cells and CD4+ T cells, which are the major contributors to lymphopenia [116,119]. This marked reduction in lymphocyte counts provides compelling evidence of suppressed adaptive immunity, though the precise mechanisms by which ASFV induces B- and T-cell depletion or apoptosis remain to be elucidated. Collectively, the combined effects of host immunosuppression and impaired B- and T-cell function result in disruption of humoral and cellular immune responses, ultimately compromising the host’s capacity to resist ASFV, leading to lethal outcomes.

## 4. Conclusions

The ASFV pandemic poses a devastating and economically significant threat to domestic pigs, prompting extensive collaborative efforts across disciplines to characterize viral proteins, understand viral pathogenesis, investigate immune responses, and develop effective vaccines and therapeutics. However, research into the molecular and immunological mechanisms of ASFV–immune system interactions is limited by the lack of easily accessible and translatable cellular and animal models, which hinders ASF disease control. Macrophages are the primary target cells for ASFV, and most research has focused on viral pathogenesis and macrophage responses, often overlooking the heterogeneity of these cells upon infection. It is crucial to recognize the complex interplay between ASFV and polarized macrophages. For instance, single-cell RNA sequencing can be employed to assess phenotypic changes in macrophages at various stages of ASFV infection, providing insight into how the virus affects macrophage polarization in vitro. Additionally, researchers can induce macrophages to polarize into M1 and M2 phenotypes in vitro and subsequently evaluate their susceptibility to ASFV to determine whether distinct macrophage subsets exhibit differential responses to the virus. Such studies could clarify how ASFV infection modulates macrophage polarization and whether these changes contribute to ASFV pathogenesis and viral dissemination.

Recent evidence indicates that both ASFV-specific antibodies and T cells may play complementary roles in combating ASFV infection. While adaptive immune responses influence virus clearance and innate immune regulation, our understanding of how protective immunity is induced by ASFV infection remains incomplete. The effectiveness of antibodies generated against ASFV proteins in neutralizing the virus is still debated, highlighting the need for precise screening of antigenic epitopes that can elicit strong neutralizing antibody responses. Additionally, more detailed mechanistic studies on T-cell-mediated immunity against ASFV in pigs are warranted. Artificial intelligence can be utilized to predict potential T-cell antigen epitopes, which can then be experimentally verified in recovered pigs from ASFV infection to identify which viral particles induce protective T-cell responses. Furthermore, analyzing transcriptomic profiles of specific T-cell subsets in the spleens of recovered pigs can reveal the signaling pathways involved in regulating their protective functions and provide insights into their antiviral mechanisms. The relationship between ASFV strains of varying virulence and the viral proteins or determinants responsible for serotype-specific adaptive immune responses in pigs has not been thoroughly characterized. By advancing our knowledge of ASFV-protective proteins, identifying precise antigenic determinants, and understanding strain diversity, we can strive to achieve an optimal balance between antibody- and T-cell-mediated immunity. This knowledge will ultimately facilitate the design and development of effective ASFV vaccines.

Most investigations into immune responses against ASFV have relied on in vitro studies, often focusing on isolated porcine alveolar macrophages, blood monocytes, or peripheral blood mononuclear cells (PBMCs) from either control or infected animals. Consequently, our understanding of the communications between innate and adaptive immune cells, as well as structural cells during ASFV infections, is relatively rudimentary. Given that ASFV antigen presentations and T-cell activation occur in situ, it is indispensable to characterize which cell types are involved in regulating immune responses against ASFV in vivo. Additionally, studying the longitudinal changes in immune cell populations and their communication will be vital for understanding the dynamics of ASFV–immune system interactions and their implications for viral transmission and pathogenesis as well as disease outcomes. To this end, it is necessary to assess the dynamic shifts in T-cell subsets in ASFV-infected pigs using techniques such as single-cell RNA sequencing, spatial transcriptome, and flow cytometry at various time points post infection. By analyzing specific subsets of innate and adaptive immune cells, along with viral gene expression within these cells, researchers can better understand the contributions of different cell types to viral transmission. Additionally, investigating the interactions between various immune cells in the context of infection could provide valuable insights into the molecular and spatiotemporal crosstalk that governs antiviral responses. This knowledge could inform the development of novel therapeutic strategies for controlling ASF.

## Figures and Tables

**Figure 1 vaccines-12-01278-f001:**
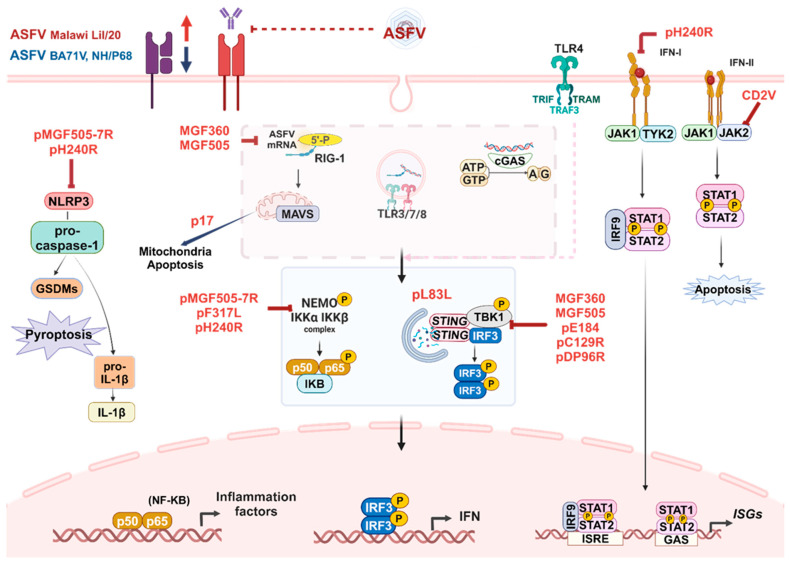
Innate immune responses against ASFV infection. Upon invading its primary target cells, macrophages, ASFV is recognized by both cell surface and intracellular pattern recognition receptors. This recognition triggers a cascade of signals transmitted through various signaling pathways, activating transcription factors such as NF-κB and IRF3, which regulate the production of IFNs and inflammatory cytokines. When IFNs bind to their receptor complex, they induce multiple downstream signaling pathways, leading to a range of biological effects. The classical STAT1/STAT2 signaling complex interacts with ISRE elements in gene promoters, resulting in the induction of numerous interferon-stimulated genes (ISGs). Additionally, STAT1 homodimers can signal inflammatory responses. The NLRP3 signaling pathway is also activated, leading to pyroptosis in infected cells. During the transcription stage of ASFV infection, ASFV proteins (highlighted in red) inhibit the production of both IFNs and inflammatory cytokines. Notably, different ASFV strains exhibit varying effects on the antigen presentation function of macrophages. Created with BioRender.com.

**Figure 2 vaccines-12-01278-f002:**
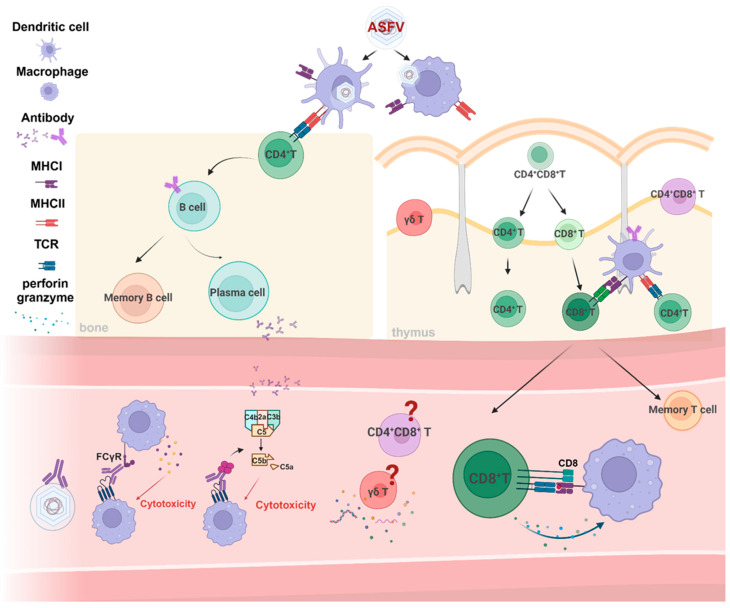
Adaptive immune responses during ASFV infection. ASFV has a significant impact on the adaptive immune response, including both humoral and cellular immunity, effectively evading immune clearance and inhibiting immune memory. B and T lymphocytes are the primary cells involved in these responses. B cells, particularly plasma cells, produce virus-specific antibodies—both neutralizing and non-neutralizing—that can neutralize antigens and trigger complement-dependent cytotoxicity as well as antibody-dependent cellular cytotoxicity. Among T cells, CD8+ T cells, including cytotoxic T lymphocytes, and CD4+ CD8+ double-positive (DP) T cells, play crucial roles. CD4+ T cells function as helper T cells, assisting in antigen presentation by antigen-presenting cells. However, ASFV infection leads to a reduction in the overall population of B cells and a decrease in CD4+ and CD8+ T cells. The proportions of DP T cells and CD4+ and CD8+ T cells can vary depending on the virulence of the ASFV strain, with the role of DP T cells during ASFV infection remaining poorly understood. Notably, pigs have a high frequency of circulating γδ T cells, which have been shown to present ASFV antigens to specific T cells in ASFV-immune pigs. However, the functional properties of these γδ T cells warrant further investigation.Created with BioRender.com.

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
