# Peer review of "Host Innate and Adaptive Immunity Against African Swine Fever Virus Infection"

_vaccines, 2024, doi:10.3390/vaccines12111278_

Round 1

Reviewer 1 Report

Comments and Suggestions for Authors

In the manuscript, the authors summarized current knowledge about the pathogenesis of ASFV and the immunological reactions of the host to infection, including both innate and adaptive immune responses. The review is well structured and informative. I think it will be useful for all researchers working in the field of ASF.

Author Response

Comment: The review is well structured and informative. I think it will be useful for all researchers working in the field of ASF.

Response: We are thankful that the reviewer thinks our review is well structured and informative, which will be useful for researchers working in the ASF field.

Reviewer 2 Report

Comments and Suggestions for Authors

The authors provide a detailed review of the literature on the innate and adaptive immunity to AFSV.  Understanding the immune response of swine to ASFV is critical to developing a vaccine or other method of protecting swine to this economic important disease.  The review in general is detailed and the authors offer some ideas on the lack of information or need for future studies on the innate and adaptive immune responses.

The article is weaken by the lack of specific suggestions or ideas on what are the next experimental steps necessary to obtain a better understanding of our gaps in the role of the innate and adaptive immune response in protecting against ASFV.  Authors could improve the manuscript by offering some future experimental paths rather than simple a concluding sentence or paragraph saying further study is needed.

Have the authors considered articles such as than by Tran et al (Transboundary and Emerging Diseases, 2021) that denotes the success of the gene deleted vaccine ASFV-G-delta177L.  There is some evidence that this vaccine does provide protection, perhaps there is some information on the innate and adaptive immune response with this vaccine?  Just a suggestion.

Overall an informative paper with good Figures.  Some review of English language is recommended.

Comments on the Quality of English Language

English quality is good but requires some review.  Some words misspelled and some use of verbs not correct.

Author Response

Comments 1: The article is weaken by the lack of specific suggestions or ideas on what are the next experimental steps necessary to obtain a better understanding of our gaps in the role of the innate and adaptive immune response in protecting against ASFV.  Authors could improve the manuscript by offering some future experimental paths rather than simple a concluding sentence or paragraph saying further study is needed.

Response 1: Thank you for your insightful comment. In response to the reviewer's suggestion, we have included additional information outlining the next experimental steps in the "Conclusion" section of the revised manuscript. These updates are highlighted in red for clarity.

Comments 2: Have the authors considered articles such as than by Tran et al (Transboundary and Emerging Diseases, 2021) that denotes the success of the gene deleted vaccine ASFV-G-delta177L.  There is some evidence that this vaccine does provide protection, perhaps there is some information on the innate and adaptive immune response with this vaccine?  Just a suggestion.

Response 2: As suggested, we have included the reference in Line 395 of revised manuscript. 

Comments 3: Overall an informative paper with good Figures.  Some review of English language is recommended.

Response 3: We are very thankful that the reviewer thinks our manuscript is "informative with good figures".  We are sorry for the oversight and have carefully gone through the manuscript to fix the typos/grammar. The updates are highlighted in red for clarity.

Reviewer 3 Report

Comments and Suggestions for Authors

Despite years of efforts to develop an ASF vaccine and several commercial vaccines licensed in Vietnam, ASF still remains one of the most significant threats to pig production worldwide. This review examines the host innate and adaptive immune responses to ASFV infection. The manuscript is well written and interesting. In my opinion, some sections lack more detail, for example, not all viral proteins involved in modulating the immune response are described. However, the authors probably did not set such tasks for themselves, since reviews on these topics have been published previously. The conclusion contains a description of the main gaps in the study of immune responses to ASFV, which should be addressed for future vaccine development efforts.

Author Response

Comments 1: The manuscript is well written and interesting. In my opinion, some sections lack more detail, for example, not all viral proteins involved in modulating the immune response are described. However, the authors probably did not set such tasks for themselves, since reviews on these topics have been published previously. The conclusion contains a description of the main gaps in the study of immune responses to ASFV, which should be addressed for future vaccine development efforts.

Response 1: We appreciate the reviewer’s comment and agree that not all viral proteins involved in modulating the immune response have been fully described. As the reviewer correctly noted, previous reviews have addressed this topic, so we have included examples of key viral proteins to summarize the potential mechanisms by which ASFV may evade macrophage responses.

Reviewer 4 Report

Comments and Suggestions for Authors

The review by Zhang et al. summarizes the current understanding of the immune responses against ASFV, encompassing both innate and adaptive immune responses, as well as insights into ASFV pathogenesis and immunosuppression. This review summarised current knowledge on ASFV interplay with macrophages in diverse polarised status, ASFV-specific antibodies and T cell-mediated immunity against ASFV. This knowledge will facilitate the design and development of effective ASFV vaccines.

The review is well written and easy to read. There are no major weaknesses. There are just few points I would like to suggest in order to improve the manuscrips:

Line 35-36. Genotype I has been recently eradicated from Sardinia, so please re-phrase this sentense.

Line 39-44. Genotype I is also circulating in China, likely due to vaccination with non licenced live attenuated vaccines.

Line 130-131. ASFV infection lead to down-regulation of TLRs as well. This can affect macrophages' ability to respond to external stimuly.

The part of macrophages/DC is quite similar to what recently reviewed by Shafer et al. 2021 (Pathogens). Please can you provide a more updated review? Please cyte and describe the results of more recents studies on ASFV interaction with macrophages in polarised status or DC.

Author Response

Comments 1: Line 35-36. Genotype I has been recently eradicated from Sardinia, so please re-phrase this sentense.

Response 1: We have revised the sentence.

Comments 2: Line 39-44. Genotype I is also circulating in China, likely due to vaccination with non licenced live attenuated vaccines.

Response 2: We thank the reviewer for pointing this out. In response, we have added this information to lines 40–41 of the revised manuscript, highlighted in red for clarity.

Comments 3: Line 130-131. ASFV infection lead to down-regulation of TLRs as well. This can affect macrophages' ability to respond to external stimuly.

Response 3: We thank the reviewer for pointing this out. In response, we have added this information to lines 260–261 of the revised manuscript, highlighted in red for clarity.

Comment 4: The part of macrophages/DC is quite similar to what recently reviewed by Shafer et al. 2021 (Pathogens). Please can you provide a more updated review? Please cyte and describe the results of more recents studies on ASFV interaction with macrophages in polarised status or DC.

Response 4: As requested, we have incorporated the findings from two studies on the interaction between African Swine Fever Virus (ASFV) and polarized macrophages (Franzoni et al., Viruses, 2022; Franzoni et al., Pathogens, 2022) into lines 179–186 of the revised manuscript. Please note that no additional studies on this topic were identified.